# Coherent States for the Isotropic and Anisotropic 2D Harmonic Oscillators

**James Moran** 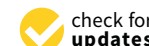 **[1,2,*] and Véronique Hussin [2,3,*]**

[1]  Département de Physique, Université de Montréal, C. P. 6128, Succ. Centre-ville,
    Montréal, QC H3C 3J7, Canada
[2]  Centre de Recherches Mathématiques, Université de Montréal, C. P. 6128, Succ. Centre-ville,
    Montréal, QC H3C 3J7, Canada
[3]  Département de Mathématiques et de Statistique, Université de Montréal, C. P. 6128, Succ. Centre-ville,
    Montréal, QC H3C 3J7, Canada
*  Correspondence: james.moran@umontreal.ca (J.M.); hussin@dms.umontreal.ca (V.H.)

**Abstract:** In this paper we introduce a new method for constructing coherent states for 2D harmonic oscillators. In particular, we focus on both the isotropic and commensurate anisotropic instances of the 2D harmonic oscillator. We define a new set of ladder operators for the 2D system as a linear combination of the $x$ and $y$ ladder operators and construct the $SU(2)$ coherent states, where these are then used as the basis of expansion for Schrödinger-type coherent states of the 2D oscillators. We discuss the uncertainty relations for the new states and study the behaviour of their probability density functions in configuration space.

**Keywords:** coherent states; harmonic oscillator; $SU(2)$ coherent states; 2D coherent states; resolution of the identity; uncertainty principle, isotropic harmonic oscillator, anisotropic harmonic oscillator

## 1. Introduction

Degeneracy in the spectrum of the Hamiltonian is one of the first problems we encounter when trying to define a new type of coherent state for the 2D oscillator. Klauder described coherent states of the hydrogen atom [1] which preserved many of the usual properties required by coherent state analysis [2]. Fox and Choi proposed the Gaussian Klauder states [3], an alternative method for producing coherent states for more general systems with degenerate spectra. An analysis of the connection between the two definitions was studied in [4].

When labeling energy eigenstates of a 2D system, $|n, m\rangle$, there exist several representations of the state space. In this paper, we present a motivation for an $SU(2)$ representation of the state space. Discussions of alternate state-space representations, as well as its application to 2D magnetism, may be found in [5,6]. When generalising beyond 2D, there exist many more state-space representations, leading to many definitions of coherent states in higher dimensions.

In this work, we aim to develop an approach for constructing coherent states for 2D oscillators in both isotropic and commensurate anisotropic settings. We aim to minimally extend the standard definitions of coherent states in the 1D setting, and we determine new properties of the constructed coherent states for the 2D system.

In the first part of the paper, we address the degeneracy in the energy spectrum by constructing non-degenerate states, the $SU(2)$ coherent states. We define a generalised ladder operator formed from a linear combination of the 1D ladder operators with complex coefficients. The $SU(2)$ coherent states are then used as a basis of expansion to describe the Schrödinger-type coherent states for the 2D system.

In the second part of the paper, we modify the $SU(2)$ coherent states according to Chen [7] to produce coherent states for the commensurate anisotropic oscillator, and we discuss the emergent properties and their correspondence to Lissajous figures in configuration space. Finally, we suggest some future directions the work can take, as well as problems that may arise in more complicated systems than the oscillator.

## 2. Coherent States of the 1D Harmonic Oscillator

The very well-known coherent states of the 1D harmonic oscillator, labelled by $z \in \mathbb{C}$, satisfy

$$a^- |z\rangle = z |z\rangle ; \tag{1}$$

$$|z\rangle = e^{(za^+ - \bar{z}a^-)} |0\rangle \equiv D(z) |0\rangle ; \tag{2}$$

$$|z\rangle = e^{-\frac{|z|^2}{2}} \sum_{n=0}^{\infty} \frac{z^n}{\sqrt{n!}} |n\rangle ; \tag{3}$$

$$\Delta \hat{x} \Delta \hat{p} = \frac{1}{2}, \forall |z\rangle \quad \text{with} \quad \Delta \hat{x} = \Delta \hat{p}. \tag{4}$$

Equations (1)–(4) describe some of the basic definitions of coherent states. These definitions were formalised by Glauber and Sudarshan [8,9], but these minimal uncertainty wave-packets were first studied by Schrödinger [10], and so we will refer to them as Schrödinger-type coherent states throughout.

Furthermore, these properties can be used to show that the states $|z\rangle$ form an over-complete basis, and they resolve the identity in the following way:

$$\int \frac{d^2z}{\pi} |z\rangle \langle z| = \sum_{n=0}^{\infty} |n\rangle \langle n| = \mathbb{I}_{\mathcal{H}}. \tag{5}$$

Here, $d^2z = d\Re z\, d\Im z$. The basis is over-complete because the states $|z\rangle$ are not orthogonal, $\langle z'|z\rangle \neq 0$. In the theory of coherent states, the resolution of the identity is often taken as a basic requirement. This allows one to use the coherent states as a basis for describing other states in the space.

## 3. The 2D Oscillator

For a 2D isotropic oscillator, we have the quantum Hamiltonian

$$\hat{H} = -\frac{1}{2}\frac{d^2}{dx^2} - \frac{1}{2}\frac{d^2}{dy^2} + \frac{1}{2}x^2 + \frac{1}{2}y^2, \tag{6}$$

where we have set $\hbar = 1$, the mass $m = 1$, and the frequency $\omega = 1$. We solve the time-independent Schrödinger equation $H |\Psi\rangle = E |\Psi\rangle$ and obtain the usual energy eigenstates (or Fock states) labelled by $|\Psi\rangle = |n, m\rangle$ with eigenvalue $E_{n,m} = n + m + 1$ and $n, m \in \mathbb{Z}^{\geq 0}$. These states may all be generated by the action of raising and lowering the operators in the following way [11]:

$$a_x^- |n, m\rangle = \sqrt{n} |n - 1, m\rangle , \; a_x^+ |n, m\rangle = \sqrt{n + 1} |n + 1, m\rangle ;$$
$$a_y^- |n, m\rangle = \sqrt{m} |n, m - 1\rangle , \; a_y^+ |n, m\rangle = \sqrt{m + 1} |n, m + 1\rangle . \tag{7}$$

In configuration space, the states $|n, m\rangle$ have the following wave-function:

$$\langle x, y|n, m\rangle = \psi_n(x)\psi_m(y) = \frac{1}{\sqrt{2^{n+m}n!m!}} \sqrt{\frac{1}{\pi}} e^{-\frac{x^2}{2} - \frac{y^2}{2}} H_n(x) H_m(y), \tag{8}$$

where $\psi_n(x) = \frac{1}{\sqrt{2^n n!}} \left(\frac{1}{\pi}\right)^{\frac{1}{4}} e^{-\frac{x^2}{2}} H_n(x)$ is the wave-function of the 1D oscillator, and $H_n(x)$ are the Hermite polynomials. For the physical position and momentum operators, $\hat{X}_i = \frac{1}{\sqrt{2}}(a_i^+ + a_i^-)$, $\hat{P}_i = \frac{1}{\sqrt{2}i}(a_i^- - a_i^+)$, respectively, and in the $i$ direction, the states $|n, m\rangle$ satisfy the following

$$(\Delta \hat{X})^2_{|n,m\rangle} = (\Delta \hat{P}_x)^2_{|n,m\rangle} = \frac{1}{2} + n; \tag{9}$$

$$(\Delta \hat{Y})^2_{|n,m\rangle} = (\Delta \hat{P}_y)^2_{|n,m\rangle} = \frac{1}{2} + m, \tag{10}$$

where $(\Delta \hat{O})^2_{|\psi\rangle} \equiv \langle\psi| \hat{O}^2 |\psi\rangle - \langle\psi| \hat{O} |\psi\rangle^2$ is the variance of the operator $\hat{O}$ in the state $|\psi\rangle$. They satisfy the Heisenberg uncertainty relation $(\Delta \hat{X})_{|n,m\rangle}(\Delta \hat{P}_x)_{|n,m\rangle} = \frac{1}{2} + n$, which grows linearly in $n$, and similarly for the $Y$ quadratures.

In what follows, we will construct two new ladder operators as linear combinations of the operators in (7) and proceed to define a single indexed Fock state for the 2D system which yields the $SU(2)$ coherent states, as well as extend the definitions in Section 2 to obtain Schrödinger-type coherent states for the 2D system.

## 4. $SU(2)$ **Coherent States**

We extend the definitions of the ladder operators presented in Section 3 to apply to the 2D oscillator. Introducing a set of states $\{|\nu\rangle\}$, and defining a new set of ladder operators through their action on the set,

$$A^- |\nu\rangle = \sqrt{\nu} |\nu - 1\rangle, \qquad A^+ |\nu\rangle = \sqrt{\nu + 1} |\nu + 1\rangle, \qquad \langle\nu|\nu\rangle = 1, \qquad \nu = 0, 1, 2, \ldots. \tag{11}$$

These states have a linear increasing spectrum, $E_\nu = \nu + 1$. We may build the states by hand, starting with the only non-degenerate state, the ground state, $|0\rangle \equiv |0, 0\rangle$, and we take simple linear combinations of the 1D ladder operators:

$$A^+_{\alpha,\beta} = \alpha\, a_x^+ \otimes \mathbb{I}_y + \mathbb{I}_x \otimes \beta\, a_y^+ ;$$
$$A^-_{\alpha,\beta} = \bar{\alpha} a_x^- \otimes \mathbb{I}_y + \mathbb{I}_x \otimes \bar{\beta} a_y^- ; \tag{12}$$
$$[A^-_{\alpha,\beta}, A^+_{\alpha,\beta}] = (|\alpha|^2 + |\beta|^2)\mathbb{I}_x \otimes \mathbb{I}_y \equiv \mathbb{I},$$

for $\alpha, \beta \in \mathbb{C}$ and $\mathbb{I}_x \otimes \mathbb{I}_y = \mathbb{I}_y \otimes \mathbb{I}_x \equiv \mathbb{I}$. Equation (12) defines the normalisation condition, $|\alpha|^2 + |\beta|^2 = 1$. Constructing the states $\{|\nu\rangle\}$ starting with the ground state gives us the following table:

**Table 1.** Construction of the states $|\nu\rangle$ using the relation $A^+ |\nu\rangle = \sqrt{\nu + 1} |\nu + 1\rangle$.

| $|\nu\rangle$ | $|n, m\rangle$ |
|---|---|
| $|0\rangle$ | $|0, 0\rangle$ |
| $|1\rangle$ | $\alpha\, |1, 0\rangle + \beta\, |0, 1\rangle$ |
| $|2\rangle$ | $\alpha^2 |2, 0\rangle + \sqrt{2}\alpha\beta |1, 1\rangle + \beta^2 |0, 2\rangle$ |
| $\vdots$ | $\vdots$ |
| $|\nu\rangle$ | $\sum_{n,m}^{n+m=\nu} \alpha^n \beta^m \sqrt{\binom{\nu}{n}} |n, m\rangle$ |

The states, $|\nu\rangle$, in Table 1 depend on $\alpha, \beta$ and may be expressed as

$$|\nu\rangle_{\alpha,\beta} = \sum_{n=0}^{\nu} \alpha^n \beta^{\nu-n} \sqrt{\binom{\nu}{n}} |n, \nu - n\rangle. \tag{13}$$

The states $|\nu\rangle_{\alpha,\beta}$ are precisely the $SU(2)$ coherent states in the Schwinger boson representation [2]. This makes sense from our construction, where the degeneracy present in the spectrum $E_{n,m}$ is an $SU(2)$ degeneracy, and so we created states which averaged out the degenerate contributions to a given $\nu$.

These states have the following orthogonality relations

$$\langle \mu|_{\gamma,\delta} \, |\nu\rangle_{\alpha,\beta} = (\bar{\gamma}\alpha + \bar{\delta}\beta)^{\nu} \delta_{\mu,\nu}, \tag{14}$$

which reduces to a more familiar relation when $\gamma = \alpha$ and $\delta = \beta$,

$$\langle \mu|_{\alpha,\beta} \, |\nu\rangle_{\alpha,\beta} = \delta_{\mu,\nu}, \tag{15}$$

using the normalization condition $|\alpha|^2 + |\beta|^2 = 1$. The states $|\nu\rangle_{\alpha,\beta}$ have the configuration space wave function expressed in terms of (8)

$$\langle x, y|\nu\rangle_{\alpha,\beta} = \sum_{n=0}^{\nu} \alpha^n \beta^{\nu-n} \sqrt{\binom{\nu}{n}} \psi_n(x)\psi_{\nu-n}(y). \tag{16}$$

In Figure 1, there are two plots of the probability density functions $\left|\langle x, y|\nu\rangle_{\alpha,\beta}\right|^2$. In the picture on the left, there is an imaginary component to the relative phase between $\alpha$ and $\beta$, and this causes the emergence of an elliptical shape to the density. Conversely, on the right, when $\alpha$ and $\beta$ are exactly in phase (or out of phase), the probability density is concentrated on a line, and the angle of the line to the $x$ axis is determined by $\tan\theta = \frac{|\beta|}{|\alpha|}$. The probability densities of the quantum $SU(2)$ coherent states mimic the spatial distribution of a classical 2D isotropic oscillator—that is, ellipses in the $(x, y)$ plane.

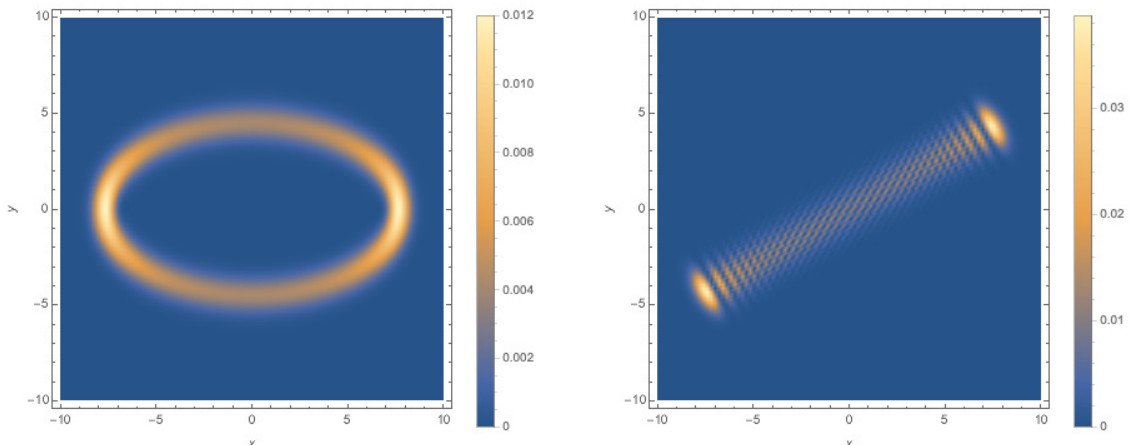

**Figure 1.** Density plots of $\left|\langle x, y|\nu\rangle_{\alpha,\beta}\right|^2$ for $\alpha = \frac{\sqrt{3}}{2}e^{i\frac{\pi}{2}}, \beta = \frac{1}{2}$ (**left**) and $\alpha = \frac{\sqrt{3}}{2}, \beta = \frac{1}{2}$ (**right**), both at $\nu = 40$.

The $SU(2)$ coherent states have the following variances for the physical position and momentum operators $\hat{X}_i = \frac{1}{\sqrt{2}}(a_i^+ + a_i^-)$, $\hat{P}_i = \frac{1}{\sqrt{2}i}(a_i^- - a_i^+)$, respectively, in the $i$ direction:

$$(\Delta \hat{X})^2_{|\nu\rangle_{\alpha,\beta}} = (\Delta \hat{P}_x)^2_{|\nu\rangle_{\alpha,\beta}} = \frac{1}{2} + |\alpha|^2\nu; \tag{17}$$

$$(\Delta \hat{Y})^2_{|\nu\rangle_{\alpha,\beta}} = (\Delta \hat{P}_y)^2_{|\nu\rangle_{\alpha,\beta}} = \frac{1}{2} + |\beta|^2\nu. \tag{18}$$

The results are essentially the same as those in (9) and (10), but they are tuned by the continuous parameters $\alpha, \beta$ introduced in (12).

## 5. Schrödinger-Type 2D Coherent States

Using the $SU(2)$ coherent states $|\nu\rangle_{\alpha,\beta}$ as a Fock basis for defining 2D coherent states in the same vein as Section 2, we write down the following:

$$|\Psi\rangle_{\alpha,\beta} = e^{-\frac{|\Psi|^2}{2}} \sum_{\nu=0}^{\infty} \frac{\Psi^\nu}{\sqrt{\nu!}} |\nu\rangle_{\alpha,\beta}. \tag{19}$$

These states have the following inner product relation:

$$\left\langle \Psi' \right|_{\gamma,\delta} |\Psi\rangle_{\alpha,\beta} = e^{-\frac{|\Psi'|^2 + |\Psi|^2}{2}} e^{\bar{\Psi}'\Psi(\bar{\gamma}\alpha + \bar{\delta}\beta)}. \tag{20}$$

Because these states are constructed so as to be analogous with the 1D definitions, we also find that they are eigenstates of the generalised lowering operator $A^-$

$$A^-_{\alpha,\beta} |\Psi\rangle_{\alpha,\beta} = \Psi |\Psi\rangle_{\alpha,\beta}. \tag{21}$$

The expansion in (19) also implies the existence of a displacement operator, as in the 1D case:

$$
\begin{aligned}
|\Psi\rangle_{\alpha,\beta} &= e^{-\frac{|\Psi|^2}{2}} \sum_{\nu=0}^{\infty} \frac{\Psi^\nu}{\sqrt{\nu!}} |\nu\rangle_{\alpha,\beta} \\
&= e^{-\frac{|\Psi|^2}{2}} \sum_{\nu=0}^{\infty} \frac{\Psi^\nu}{\sqrt{\nu!}} \frac{A^{+\,\nu}_{\alpha\beta}}{\sqrt{\nu!}} |0\rangle_{\alpha,\beta} \\
&= e^{-\frac{|\Psi|^2}{2} + \Psi A^+_{\alpha\beta}} |0\rangle_{\alpha,\beta} \equiv D(\Psi) |0\rangle_{\alpha,\beta}.
\end{aligned}
\tag{22}
$$

A Baker-Campbell-Haussdorf identity, along with the annihilation of the 2D vacuum, $A^-_{\alpha,\beta} |0\rangle_{\alpha,\beta} = 0$ allows us to rewrite $D(\Psi)$ in the following way:

$$
\begin{aligned}
D(\Psi) &= e^{\Psi A^+_{\alpha,\beta} - \bar{\Psi} A^-_{\alpha,\beta}} \\
&= e^{(\alpha\Psi a^+_x - \bar{\alpha}\bar{\Psi} a^-_x) + (\beta\Psi a^+_y - \bar{\beta}\bar{\Psi} a^-_y)} \\
&= D_x(\alpha\Psi) D_y(\beta\Psi),
\end{aligned}
\tag{23}
$$

where we have split $D(\Psi)$ into operators acting on $x$ and $y$ independently. The Schrödinger-type coherent states then factorise into two uncoupled 1D coherent states, $|\alpha\,\Psi\rangle_x \otimes |\beta\,\Psi\rangle_y$.

The Schrödinger-type coherent states represent an infinite sum of the elliptical, or $SU(2)$ coherent states established previously, with a Poissonian probability of being in a state $|\mu\rangle_{\alpha,\beta}$ given by

$$\left| \langle \mu |_{\alpha,\beta} |\Psi\rangle_{\alpha,\beta} \right|^2 = e^{-|\Psi|^2} \frac{|\Psi|^{2\mu}}{\mu!}, \tag{24}$$

analogous to the 1D coherent states, $|\langle n|z\rangle|^2 = e^{-|z|^2} \frac{|z|^{2n}}{n!}$.

It is clear from the factorisation of the displacement operator (23) that the wave-function of the Schrödinger-type coherent states must also factorise into the product of two 1D coherent state wave-functions. Using the general form of the 1D coherent state wave-function [2], we get the position representation of the 2D Schrödinger-type coherent states:

$$\langle x, y | \Psi\rangle_{\alpha,\beta} = \frac{1}{\sqrt{\pi}} \exp\left( -\frac{1}{2}[(x - \sqrt{2}\,\text{Re}(\alpha\Psi))^2 + (y - \sqrt{2}\,\text{Re}(\beta\Psi))^2] \right) e^{\left(i\sqrt{2}[x\,\text{Im}(\alpha\Psi) + y\,\text{Im}(\beta\Psi)]\right)}. \tag{25}$$

In Figure 2, we see the probability densities $\left|\langle x, y|\Psi\rangle_{\alpha,\beta}\right|^2$ are Gaussian in the $(x, y)$ plane. The peak of the probability density is located at the coordinates $(x, y) = (\sqrt{2}\,\text{Re}(\alpha\Psi), \sqrt{2}\,\text{Re}(\beta\Psi))$.

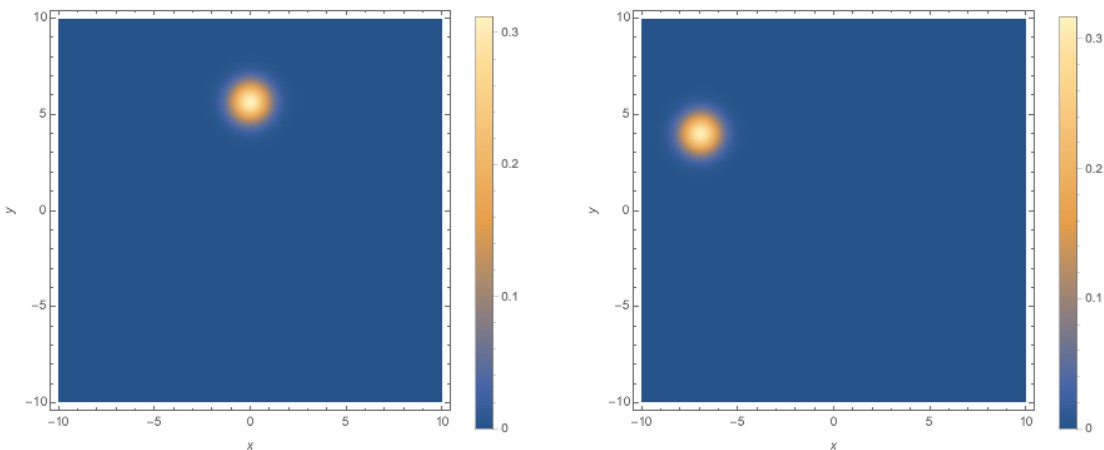

**Figure 2.** Density plots of $\left|\langle x, y|\Psi\rangle_{\alpha,\beta}\right|^2$ for $\Psi = 8, \alpha = \frac{\sqrt{3}}{2}e^{i\frac{\pi}{2}}, \beta = \frac{1}{2}$ (**left**) and $\Psi = 8e^{i\frac{\pi}{4}}, \alpha = \frac{\sqrt{3}}{2}e^{i\frac{\pi}{2}}, \beta = \frac{1}{2}$ (**right**).

The Schrödinger-type 2D isotropic coherent states are minimal uncertainty states in both $x$ and $y$, and this follows from the factorisation of the displacement operator,

$$(\Delta\hat{X})_{|\Psi\rangle_{\alpha,\beta}}(\Delta\,\hat{P}_x)_{|\Psi\rangle_{\alpha,\beta}} = \frac{1}{2}, \qquad (\Delta\hat{X})_{|\Psi\rangle_{\alpha,\beta}} = (\Delta\,\hat{P}_x)_{|\Psi\rangle_{\alpha,\beta}}; \tag{26}$$

$$(\Delta\hat{Y})_{|\Psi\rangle_{\alpha,\beta}}(\Delta\,\hat{P}_y)_{|\Psi\rangle_{\alpha,\beta}} = \frac{1}{2}, \qquad (\Delta\hat{Y})_{|\Psi\rangle_{\alpha,\beta}} = (\Delta\,\hat{P}_y)_{|\Psi\rangle_{\alpha,\beta}}. \tag{27}$$

## 6. Resolution of the Identity

The $SU(2)$ coherent states resolve the identity in the following way:

$$\frac{\nu + 1}{\pi^2} \int_{S^3} d^2\alpha \, d^2\beta \, \delta(|\alpha|^2 + |\beta|^2 - 1) \, |\nu\rangle_{\alpha,\beta} \, \langle\nu|_{\alpha,\beta} = \mathbb{I}_\nu, \tag{28}$$

where $\mathbb{I}_\nu$ is the identity operator for the states $\{|\nu\rangle_{\alpha,\beta}\}$—in other words, the sum of the projectors onto states with a total occupation number of $n + m = \nu$—for example, $\mathbb{I}_2 = |2,0\rangle\langle 2,0| + |1,1\rangle\langle 1,1| + |0,2\rangle\langle 0,2|$.

We retrieved the identity operator for the entire Hilbert space by summing over $\nu$

$$\sum_{\nu=0}^{\infty}\left(\frac{\nu+1}{\pi^2}\int_{S^3} d^2\alpha \, d^2\beta \, \delta(|\alpha|^2 + |\beta|^2 - 1)\,|\nu\rangle_{\alpha,\beta}\,\langle\nu|_{\alpha,\beta}\right) = \sum_{n=0}^{\infty}\sum_{m=0}^{\infty}|n,m\rangle\langle n,m| = \mathbb{I}_{\mathcal{H}}. \tag{29}$$

The resolution of the identity allowed us to express any other state in the Hilbert space in terms of the states $\{|\nu\rangle_{\alpha,\beta}\}$. The energy eigenstates were then given by

$$|n,m\rangle = \sum_{\nu=0}^{\infty}\left\{\frac{\nu+1}{\pi^2}\int_{S^3} d^2\alpha \, d^2\beta \, \delta(|\alpha|^2 + |\beta|^2 - 1)\sqrt{\binom{\nu}{n}}\bar{\alpha}^n\bar{\beta}^m\,|\nu\rangle_{\alpha,\beta}\right\}. \tag{30}$$

The Schrödinger-type 2D coherent states resolve the identity with a slightly modified measure. It is insufficient to combine the measures used for the 1D coherent states and $SU(2)$ coherent states in Equations (5) and (28), where doing so, we would obtain

$$\frac{1}{\pi^2} \int_{S^3} d^2\alpha \ d^2\beta \ \delta(|\alpha|^2 + |\beta|^2 - 1) \int_{\mathbb{C}} \frac{d^2\Psi}{\pi} |\Psi\rangle_{\alpha,\beta} \langle\Psi|_{\alpha,\beta} = \sum_{\nu=0}^{\infty} \frac{\mathbb{I}_\nu}{\nu+1} = \sum_{n=0}^{\infty} \sum_{m=0}^{\infty} \frac{|n,m\rangle \langle n,m|}{n+m+1} \not\propto \mathbb{I}_{\mathcal{H}}. \quad (31)$$

However, the identity operator for the full Hilbert space can be retrieved by the inclusion of $|\Psi|^2$ into the measure as follows:

$$\frac{1}{\pi^2} \int_{S^3} d^2\alpha \ d^2\beta \ \delta(|\alpha|^2 + |\beta|^2 - 1) \int_{\mathbb{C}} \frac{d^2\Psi}{\pi} |\Psi|^2 |\Psi\rangle_{\alpha,\beta} \langle\Psi|_{\alpha,\beta} = \mathbb{I}_{\mathcal{H}}, \quad (32)$$

thus, the Schrödinger-type coherent states for the 2D oscillator represent an over-complete basis for the full Hilbert space of the 2D oscillator. The resolution of the identity means the states could have some application in 2D coherent state quantization [2].

## 7. Commensurate Anisotropic $SU(2)$ Coherent States

In order to generalise coherent states to the commensurate anisotropic oscillator, we introduce two integers, $p, q$, in the Hamiltonian as

$$\begin{aligned} \hat{H} &= -\frac{1}{2}\frac{d^2}{dx^2} - \frac{1}{2}\frac{d^2}{dy^2} + \frac{1}{2}\omega_x^2 x^2 + \frac{1}{2}\omega_y^2 y^2 \\ &= -\frac{1}{2}\frac{d^2}{dx^2} - \frac{1}{2}\frac{d^2}{dy^2} + \frac{p^2}{2}\omega^2 x^2 + \frac{q^2}{2}\omega^2 y^2, \end{aligned} \quad (33)$$

where the frequencies are related by $\omega_x = p\omega$ and $\omega_y = q\omega$, and the ratio, $\frac{p}{q}$, represents the ratio of the two frequencies, $\frac{\omega_x}{\omega_y}$. Without loss of generality, we will set the common frequency $\omega = 1$ in what follows and choose $p, q$ such that they are relative prime integers. A hypothesis made by Chen [12] says that the integers $p, q$ enter the quantum $SU(2)$ coherent states in the following way:

$$|\nu\rangle_{\alpha,\beta}^{p,q} = \sum_{n=0}^{\nu} \alpha^n \beta^{\nu-n} \sqrt{\binom{\nu}{n}} |pn, q(\nu-n)\rangle, \quad (34)$$

where the states are normalised in the usual way: $\langle\nu|_{\alpha,\beta}^{p,q} |\nu\rangle_{\alpha,\beta}^{p,q} = 1$ and $|\alpha|^2 + |\beta|^2 = 1$.

Chen's hypothesis (34) suitably addresses the extension of our construction to the commensurate anisotropic oscillator. Energy eigenstates of (33) have eigenvalues $E_{n,m} = p\left(n + \frac{1}{2}\right) + q\left(m + \frac{1}{2}\right)$, which do not have the same degenerate structure as in the isotropic case where $p = q = 1$, and instead we are considering a superposition of states $|pn, qm\rangle$ such that $n + m = \nu$ for given $p, q$.

The energy eigenvalues of the states $|\nu\rangle_{\alpha,\beta}^{p,q}$ may be calculated from

$$\begin{aligned} \langle\nu|_{\alpha,\beta}^{p,q} a_x^+ a_x^- + a_y^+ a_y^- + 1 |\nu\rangle_{\alpha,\beta}^{p,q} &= (p-q)\left(\sum_{n=0}^{\nu} |\alpha|^{2n}|\beta|^{2(\nu-n)}\binom{\nu}{n}n\right) + q\nu + 1 \\ &= (p-q)|\alpha|^2\nu + q\nu + 1 \\ &= p|\alpha|^2\nu + q|\beta|^2\nu + 1 \\ &\equiv E_\nu^{p,q}, \end{aligned} \quad (35)$$

which was computed by observing that

$$\frac{\partial}{\partial |\alpha|^2} \sum_{n=0}^{\nu} |\alpha|^{2n} |\beta|^{2(\nu-n)} \binom{\nu}{n} = \frac{\partial}{\partial |\alpha|^2} (|\alpha|^2 + |\beta|^2)^\nu, \tag{36}$$

yielding

$$\sum_{n=0}^{\nu} |\alpha|^{2(n-1)} |\beta|^{2(\nu-n)} \binom{\nu}{n} n = \nu(|\alpha|^2 + |\beta|^2)^{\nu-1} = \nu. \tag{37}$$

The states $|\nu\rangle_{\alpha,\beta}^{p,q}$ correspond to Lissajous-type probability densities in configuration space, a feature present in the classical spatial distribution of an anisotropic oscillator with commensurate frequencies [7,13].

In Figure 3, we have two types of Lissajous figures, where on the left is a closed figure and on the right, an open figure. The frequency ratio $\frac{p}{q}$ determines the type of Lissajous figure, and the relative phase between $\alpha$ and $\beta$ deforms the figures such that when they are completely in (or out of) phase, the figure is open, and when there is an imaginary component to the relative phase, the figure is closed. Tables of Lissajous figures corresponding to different choices of $p$ and $q$ can be found in [14]. The correspondence of the quantum probability densities to the classical spatial distribution of a 2D commensurate anisotropic oscillator confirms Chen's definition as a suitable description of coherent states.

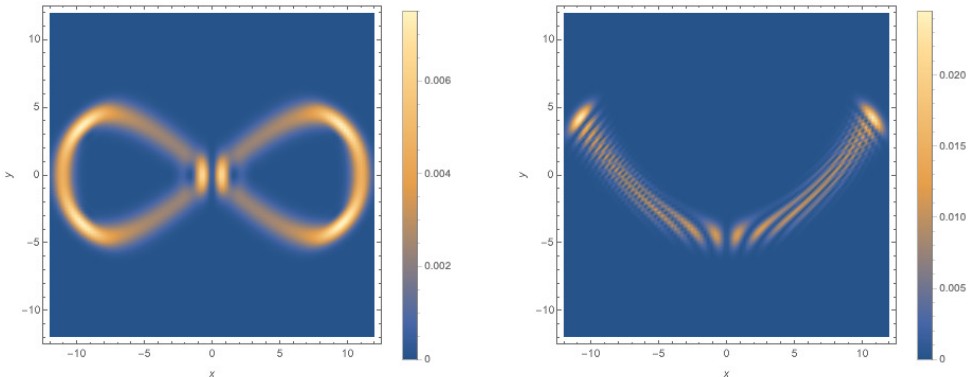

**Figure 3.** Density plots of $\left|\langle x,y|\nu\rangle_{\alpha,\beta}^{p,q}\right|^2$ for $\alpha = \frac{\sqrt{3}}{2} e^{i\frac{\pi}{2}}, \beta = \frac{1}{2}$ (**left**) and $\alpha = \frac{\sqrt{3}}{2}, \beta = \frac{1}{2}$ (**right**) for $p = 2, q = 1$ at $\nu = 40$.

The commensurate anisotropic $SU(2)$ coherent states have slightly modified variances compared with the isotropic case

$$(\Delta \hat{X})^2_{|\nu\rangle_{\alpha,\beta}^{p,q}} = (\Delta \hat{P}_x)^2_{|\nu\rangle_{\alpha,\beta}^{p,q}} = \frac{1}{2} + |\alpha|^2 p\nu; \tag{38}$$

$$(\Delta \hat{Y})^2_{|\nu\rangle_{\alpha,\beta}^{p,q}} = (\Delta \hat{P}_y)^2_{|\nu\rangle_{\alpha,\beta}^{p,q}} = \frac{1}{2} + |\beta|^2 q\nu. \tag{39}$$

## 8. Commensurate Anisotropic 2D Schrödinger-Type Coherent States

As with the isotropic case, we can build 2D Schrödinger-type coherent states using the commensurate anisotropic $SU(2)$ coherent states as a basis, defining the states $|\Psi\rangle_{\alpha,\beta}^{p,q}$

$$|\Psi\rangle_{\alpha,\beta}^{p,q} = e^{-\frac{|\Psi|^2}{2}} \sum_{\nu=0}^{\infty} \frac{\Psi^{\nu}}{\sqrt{\nu!}} |\nu\rangle_{\alpha,\beta}^{p,q}. \tag{40}$$

These Schrödinger-type coherent states are normalised $\langle\Psi|_{\alpha,\beta}^{p,q} |\Psi\rangle_{\alpha,\beta}^{p,q} = 1$ with inner product

$$\langle\Psi'|_{\alpha,\beta}^{p,q} |\Psi\rangle_{\alpha,\beta}^{p,q} = e^{-\frac{|\Psi'|^2+|\Psi|^2}{2}} e^{\bar{\Psi}'\Psi}. \tag{41}$$

Similarly to the isotropic case, (40) may be interpreted as the infinite sum of commensurate anisotropic $SU(2)$ coherent states, determined by $p, q$, with a probability of being in a given coherent state, $|\mu\rangle_{\alpha,\beta}^{p,q}$, given by

$$\left|\langle\mu|_{\alpha,\beta}^{p,q} |\Psi\rangle_{\alpha,\beta}^{p,q}\right|^2 = e^{-|\Psi|^2} \frac{|\Psi|^{2\mu}}{\mu!}. \tag{42}$$

Figures 4 and 5 show four density plots for the probability density of the commensurate anisotropic 2D Schrödinger-type coherent states. We have finitely used many terms in the expansion of $|\Psi\rangle_{\alpha,\beta}^{p,q}$, and so we can see the emergence of localisation, but the pictured graphs are not properly normalised as a result. An interesting difference between the isotropic and commensurate anisotropic Schrödinger-type coherent states is that for certain values of $(\alpha, \beta, \Psi, p, q)$, the probability density can localise onto two or more separate points. This can be seen clearly in the left-most image in Figure 5, unlike the isotropic Schrödinger states which were seen to have Gaussian probability distributions in configuration space with a single maximum.

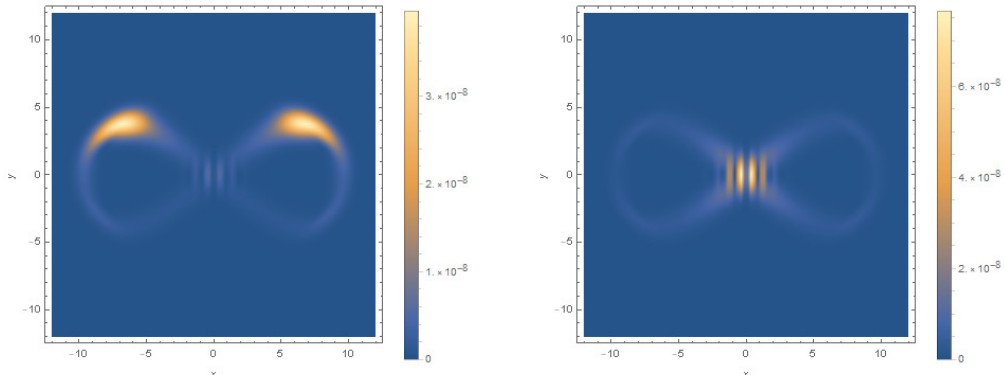

**Figure 4.** Density plots of $\left|\langle x,y|\Psi\rangle_{\alpha,\beta}^{p,q}\right|^2$ for $\Psi = 8, \alpha = \frac{\sqrt{3}}{2}e^{i\frac{\pi}{2}}, \beta = \frac{1}{2}$ (**left**) and $\Psi = 8e^{i\frac{\pi}{2}}, \alpha = \frac{\sqrt{3}}{2}e^{i\frac{\pi}{2}}, \beta = \frac{1}{2}$ (**right**), with $p = 2, q = 1$ in both instances. Thirty terms are kept in the expansion of $|\Psi\rangle_{\alpha,\beta}^{p,q}$. We see the emergence of localisation onto parts of the $SU(2)$ coherent state used in the expansion.

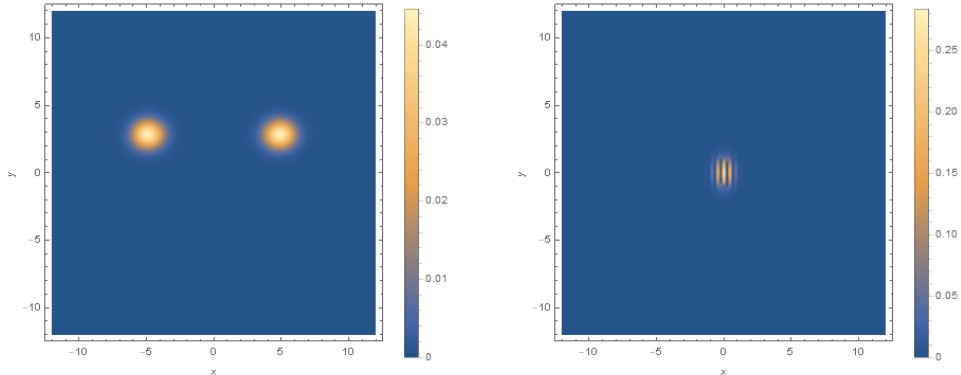

**Figure 5.** Density plots of $\left|\langle x,y|\Psi\rangle_{\alpha,\beta}^{p,q}\right|^2$ for $\Psi = 4, \alpha = \frac{\sqrt{3}}{2}e^{i\frac{\pi}{2}}, \beta = \frac{1}{2}$ (**left**) and $\Psi = 4e^{i\frac{\pi}{2}}, \alpha = \frac{\sqrt{3}}{2}e^{i\frac{\pi}{2}}, \beta = \frac{1}{2}$ (**right**), with $p = 2, q = 1$ in both instances. Thirty terms are kept in the expansion of $|\Psi\rangle_{\alpha,\beta}^{p,q}$.

In the right-most density plot in Figure 5 there is good localisation, but the probability distribution is fringed around the origin, and this behaviour differs from the isotropic counterparts. The graphs in Figure 4 are clearly far from normalisation (because larger $\Psi$ was used), but they demonstrate how the first few terms in the expansion of $|\Psi\rangle_{\alpha,\beta}^{p,q}$ begin to localise onto the Lissajous figure. The parameters $(\alpha, \beta, p, q)$ determine the topology of the Lissajous figure, as described in Section 7, while $\arg \Psi$ controls the points on the Lissajous figure where the probability density will concentrate.

## 9. Conclusions

In this paper we have described a method for constructing coherent states for the 2D oscillator, which relies on using the minimal set of definitions used to describe the coherent states of the 1D oscillator. We found that most of the properties of the 1D coherent states were also present in their 2D isotropic Schrödinger-type counterparts: minimisation of the uncertainty principle, existence of a displacement operator, eigenstates of an annihilation operator, and correspondence to classical dynamics. A suitable measure was also found for the resolution of the identity.

Using the hypothesis of Chen, we generalised these results to the commensurate anisotropic 2D harmonic oscillator and found that their probability densities corresponded to Lissajous orbits. It is not clear at present how these results can be extended to the non-commensurate case. The relative prime integers $p, q$ enter the $SU(2)$ coherent states in a very natural way, but it seems that a different formalism altogether would be required when dealing with non-commensurable $\omega_x, \omega_y$, where classically this would correspond to quasi-periodicity [15].

As an outlook, it would be interesting to obtain detailed results on the variances of the physical quadratures in the commensurate anisotropic Schrödinger-type coherent states. We were able to assess the localisation of the probability densities, but they lacked accurate numerical values as a result of using a finite number of terms in the expansion of $|\Psi\rangle_{\alpha,\beta}^{p,q}$. A further consideration would be to define a squeezing operator with the generalised ladder operators $A^-$ and $A^+$, analogously to the 1D squeezing operator, $S(\Xi) = e^{\frac{\Xi}{2}(A^+)^2 - \frac{\bar{\Xi}}{2}(A^-)^2}$. When acting on the ground state with this operator to produce a 2D squeezed vacuum $S(\Xi)|0\rangle$, we obtained non-trivial interactions between the $x$ and $y$ oscillators due to the bilinear terms appearing in the exponent. A two-mode-like squeezing between $x$ and $y$ modes was found to arise.

Finally, this method could perhaps be used to describe coherent states for degenerate systems other than the harmonic oscillator, where the 2D oscillator is the simplest example of a degenerate 2D spectrum, and the next simplest example would be the particle in a 2D box. Work has been done on defining coherent states with degenerate spectra by Fox and Choi [3], and the example of the particle in a 2D box was looked at in [4]. The extension of our framework is not extremely

straightforward; however, the spectrum of the particle in a box goes as $n^2 + m^2$, which contains non-algebraic degeneracies (such as $1^2 + 7^2 = 5^2 + 5^2$) and would require more careful thought when counting states in a given degenerate subgroup $|\nu\rangle$.

**Author Contributions:** Each author contributed equally to this article.

**Funding:** This research received no external funding.

**Acknowledgments:** V.H. acknowledges the support of research grants from NSERC of Canada.

**Conflicts of Interest:** The authors declare no conflict of interest.

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
