# Peer review of "Coherent States for the Isotropic and Anisotropic 2D Harmonic Oscillators"

_quantumrep, doi:10.3390/quantum1020023_

Round 1

Reviewer 1 Report

This paper is addressing an interesting problem.

The  paper cannot  be understood without relying on reference (5),moreover only ten lines are devoted to the 2D anisotropic case of Schrödinger type coherent states ,which is the appealing aspect in the title.Consider,in this respect ,the role of SU(2) coherent states .

In the construction of A and A* ,formula (12),it would be appropriate ,to label the operators by means of  alpha and beta.By changing their values one gets a different realization of the Heisenberg-Weyl algebra.Their commutation relations should also be given and the normalization of the vector (alpha,beta)would follow from the requirement of the standard commutations relations.

When it comes to section "7.Anisotropic  SU(2) coherent states" ,they should specify that they are dealing with the ratio of the two frequencies which is the ratio of two relatively prime numbers.They do not consider the case of an irrational ratio of the two frequencies.

At the classical level,when the ratio of the two frequencies is irrationa,the corresponding orbit would be not closed ,its closure would be the two-dimensional torus defined by the level set of the two energies,it means the system would not allow for further constants of the motion.What happens to the degeneracy of the energy levels ?

I do not expect that the authors tackle this problem ,however they should at least clearly say that they are limiting their considerations to sytems whose classical orbits would be closed.

A further comment is in order : on page two ,line 32,they correctly say that

"These definitions were formalized by Glauber and Sudarshan [6],..."

However in reference [6] only the paper of Glauber is quoted ,the Physical Review Letter by Sudarshan is not quoted .

My overall recommendation is to publish the paper ,but the authors should make an effort to improve the presentation.

Author Response

Thank you for the feedback, we have implemented the suggestions:

We have added more detail to the anisotropic Schroedinger coherent states, though exact results for the dispersions are still a work in progress.

We added appropriate subscripts to the operators A^+, A^-

We have made it more clear that we are dealing with commensurable frequencies in the anisotropic setting, and have suggested that different definitions would be required for finding correspondence with classical non-commensurate oscillators

We have also added the reference for Sudarshan, that was an unfortunate oversight.

We have also addressed other parts of the paper to improve the overall presentation: the exact wavefunction for the 2d isotropic Schroedinger coherent states, more results for the resolution of the identity, and modified the introduction and conclusion to reflect these changes.

Reviewer 2 Report

In the submitted manuscript, the authors propose the construction of coherent states for the two-dimensional (2D) isotropic and anisotropic harmonic oscillators based on the definitions of coherent states in the one-dimensional (1D) case and analyze their properties.

In my opinion, this work is relevant because extends the concept of coherent states from 1D to 2D for the harmonic oscillator in a clear and well-justified way.

I would recommend the publication of the mansucript after the following issues are addressed:

Please change don't -> do not , in the Introduction, line 11. In all figures, please include a legend to identify the colors of the plots with numerical values and enlarge the axes labels if it is possible.

Author Response

Thank you for your feedback.

We have made several changes including adding legends to all density plots, as well as correcting the language throughout, e.g. don't -> do not etc

We have also addressed other parts of the paper to improve the overall presentation: the exact wavefunction for the 2d isotropic Schroedinger coherent states, more results for the resolution of the identity, and modified the introduction and conclusion to reflect these changes.

Reviewer 3 Report

The manuscript aims at constructing coherent states for both the isotropic and anisotropic 2D harmonic oscillators. For this purpose the authors eliminate the degeneracy of the Fock states by constructing the non-degenerate basis of SU(2) coherent states, characterized by the total number of excitations. This new basis is then used to define the Schrödinger-type 2D coherent states in a way that mimics the definition of the usual coherent states in terms of the (usual) Fock basis. Some properties of these Schrödinger-type states are analyzed, finding that they satisfy some of the properties exhibited by (usual) coherent states but fail in satisfying others, as the completeness property. The analysis regarding the anisotropic case is similar to the isotropic one, but involving “generalized” SU(2) coherent states following the work of Chen, and focuses on the Lissajous figures exhibited by the probability density in configuration space.  

The manuscript is well-written, and the findings may be useful in studying other interesting cases. The results presented seem correct, though most of the expressions are simply stated without a clue of the explicit derivation.  

Some comments/questions which would serve to improve the paper are the following:

The introduction of the paper should be improved, making clear the main differences/advantages of the present approach when compared with previous attempts of constructing 2D coherent states. Also, the phrase “we determine new properties as well as missing properties in our definitions of coherent states” seemed somewhat obscure to me. Which are those “new” properties? And the missing ones? Do the authors refer here to those properties of usual coherent states that are not satisfied by the Schrödinger-type coherent states?   Below Eq. (8) it is convenient to recall that H_n stands for the Hermite polynomials. In lines 54-55 it is said that “the probability densities … mimick the motion of a classical 2D isotropic oscillator, that is, elliptical motion in the plane”. Since the probability density in configuration space says nothing about the motion, I think it is more appropriate to avoid the use of the word `motion’, and write instead “the probability densities … mimic the classical 2D isotropic oscillator spatial distribution.”   When introducing the displacement operator D(\Psi), below Eq. (21), it is not clear from the presentation that the state |\Psi>_{\alpha, \beta} is indeed obtained by applying D(\Psi) to the ground state |\nu=0>. This should be demonstrated; otherwise the analogy with Eq. (2) is missing.  In relation with the above point, the factorization (22) of the displacement operator implies that the state |\Psi>_{\alpha, \beta} must be identified with the tensor product of two 1D coherent states, namely |z_x>=|\alpha \Psi> and |z_y>=|\beta \Psi>. Therefore, in line 63 the term “these” in the phrase “It is important to note that these represent…” refers precisely to the Schrödinger-type states, right? In that case, I find confusing the second line of the Introduction, stating that “…new type of coherent states for the 2D oscillator, states which don’t simply factorise into the product of two coherent states”. So, the constructed states do factorize, yet the 2D coherente states do not factorize? Clarification of this confusion seems necessary in the text.  As a consequence of the aforementioned factorization, it is said that the Schrödinger-type states constitute a subset of typical (product) 2D coherent states, which ultimately explains why they do not represent a complete basis of the full Hilbert space. Can the authors add some comments regarding the limitations of this result? How bad is to deal with coherent states that do not constitute a basis of the corresponding Hilbert space? Are there any physical consequences of this? The frequencies in the Hamiltonian (30) should be squared. The construction of the coherent states for the anisotropic case rests fundamentally on a hypothesis made by Chen, yet nothing is said regarding the nature or support of such hypothesis. The authors should at least include some lines explaining its origin and meaning. 

Finally, I consider that the manuscript can be accepted for publication once the authors address the above points. 

Author Response

Thank you for your detailed feedback.

We have addressed the ambiguous wording in the introduction, we hope it is clearer.

We have referenced that H_n(x) refer to the Hermite polynomials. As well we recognise that it was poor wording to say the states mimic the motion of the classical system -> this has been changed to spatial distribution.

We have included the equivalence between the displacement operator and its expansion for clarity.

We have made the factorisation of the states clearer in the text, specifying that it is exactly |\alpha \Psi> \otimes |\beta \Psi>

We have since realised that the states do resolve the identity with respect to a modified measure, namely, including the term |\Psi|^2 into the measure. We hope this is clearer throughout the text, these states do in fact constitute a complete basis in the Hilbert space.

We have also added comments explaining why Chen's hypothesis for the anisotropic SU(2) coherent states works well within our framework.

Reviewer 4 Report

In this work the authors provide a route to construct non-degenerate coherent states for the two-dimensional isotropic harmonic oscillator, which satisfy analogous (though generalized) relations as the usual one-dimensional coherent states. The resulting states happen to depend only on two parameters, \alpha and \beta, which somehow determine the topology displayed by the density of the corresponding state, ranging from seemingly classical-like closed curves to patterns with strong interferential traits. These topologies are related to the classical Lissajous figures. The work is completed with an extension to (two-dimensional) anisotropic harmonic oscillators with commensurate frequencies, obtaining similar results, although depending on the ratio between the two integers associated with each frequency.

My opinion is that the introduction of this coherent state definitions makes a potential contribution of interest to the field and therefore the work deserves consideration for publication in the current form. Although the authors discuss some alternative extensions of the methodology in the last section, if it is familiar to them, maybe they could also comment on whether it could also be extended to the case of anisotropic harmonic oscillators characterized by incommesurate frequencies, due to the ergodicity of their classical counterparts and its relationship to (classical) chaotic dynamics when there is presence of nonlinear coupling terms.

Author Response

Thank you for your feedback.

We have implemented a number of changes through, trying to add detail where it was lacking. The most major revisions are:a new result for the resolution of the identity with respect to a modified measure, legends added to density plots. modifications were made to the introduction and conclusion to make motivations and future works clearer.

Round 2

Reviewer 3 Report

In this revised version the authors have satisfactorily taken into account my previous comments, so I recommend the manuscript for publication. One minor point: Above Eq. (31) it says "The Schrödinger-type 2D coherent resolve the..." and its should say "The Schrödinger-type 2D coherent STATES resolve the..."